# Computer-Assisted Intraoperative Navigation in Pediatric Head and Neck Surgical Oncology: A Single-Center Case Series and Scoping Review of the Literature

**DOI:** 10.3390/cancers18010154

**Published:** 2026-01-01

**Authors:** Jordan Whittles, Ajay Bharathan, Shannon Hall, James Baumgartner, Joseph Lopez

**Affiliations:** 1Loma Linda University School of Medicine, Loma Linda, CA 92350, USA; 2Division of Pediatric Head & Neck Surgery, Department of Children’s Surgery, AdventHealth for Children, 401 N Mills Ave, Suite C, Orlando, FL 32803, USA; 3University of Central Florida College of Medicine, Orlando, FL 32827, USA; 4Department of General Surgery, AdventHealth Orlando, Orlando, FL 32803, USA; 5Division of Pediatric Neurosurgery, Department of Children’s Surgery, AdventHealth for Children, Orlando, FL 32803, USA

**Keywords:** pediatric plastic surgery, pediatric surgical oncology, pediatric head and neck cancer, intraoperative navigation, scoping review

## Abstract

Our review and institutional experience highlight the emerging yet meaningful role of intraoperative navigation (iNav) in pediatric head and neck surgical oncology. Across the twenty-seven cases identified in the literature and the five cases from our center, iNav consistently supported surgeons in identifying patient-specific anatomy, delineating tumor margins, and confirming the depth and adequacy of resection. Although the available evidence is largely limited to lower-level studies, the authors uniformly endorsed iNav as a useful adjunct, particularly in anatomically complex or previously operated fields, despite manageable limitations related to registration and tissue manipulation. Our single-center experience to date further demonstrates a complete rate of negative-margin resections with its use. Taken together, these findings reinforce that iNav enhances operative confidence and precision in pediatric head and neck cancer surgery, supporting its continued integration as a tool that may help balance oncologic completeness with minimizing surgical morbidity.

## 1. Introduction

Historically, surgeons have been dependent on general anatomical mastery to actualize operative plans. However, within the last two decades, technological advancements have allowed for the development of individualized anatomical models, which may be computerized or physical, to provide the basis for personalized surgical treatment. These patient-specified models allow surgeons to prepare unique procedures with preoperative planning, identify the optimal approach, and virtually test the efficacy of their planned interventions [1,2].

However, despite these preoperative technological advancements (i.e., computer-assisted surgical planning (CASP), computer-assisted design and manufacturing (CAD/CAM), etc.), there remains a conceptual gap. The intraoperative actualization of a surgical plan is confronted by the physical reality and complexity of the patient or problem, where the surgical approach, landmarks, and trajectories may not be so easily identified and mapped to the intended plan. Ideally, the predefined surgical modeling could be seamlessly integrated into the operative environment and workflow. A spectrum of technologies may bridge this gap between the surgical intention and operative actualization. On one end of this spectrum lie the reference-comparative approaches. The most basic of these approaches is the use of intraoperative imaging [3,4]. Since imaging data is the basis of every individualized surgical plan, collecting another scan concurrently with or immediately post-intervention allows for the confirmation of successful plan implementation or the opportunity for further adjustment or intervention. However, its limitations are obvious: extended operative and anesthesia time, potentially additional radiation exposure (with CT or fluoroscopy), complex surgical workflows, and extensive infrastructure and equipment requirements. In addition, the resultant imaging is not necessarily reintegrated into the preoperative planning environment, leaving the imaging received as a proximate estimate of intended effect.

Another comparative approach entails the use of computer-assisted manufacturing to create three-dimensional personalized anatomical models [5,6,7,8,9]. These stereolithic models may be printed using a high-resolution 3D printer and sterilized for use in the operating room, which allows for differentiation of the relevant surgical landmarks based on multicolored printing or detachable elements. Additionally, multiple models from various stages of the surgical plan may be used for comparison and monitoring of intraoperative progression. Although model-based comparison is a helpful surgical adjunct, it is still limited in its ability to bridge the conceptual gap; the surgeon must still traverse the spatial ambiguity of the operative field to superimpose the surgical plan from the model.

On the other end of the spectrum are the integrative approaches. The quintessential example of this integration would be a high-fidelity, mixed/augmented-reality system, which could directly import or superimpose visual guidance into the surgeon’s field of view for the ideal surgical approach. These methods are in development both preclinically and clinically, yet will remain unvalidated for the foreseeable future [10]. However, intraoperative navigation (iNav) represents an integrated approach which allows for the direct comparison of intraoperative spatial datapoints to the preoperative surgical plan. The most common varieties of iNav function according to the optical physics principle of parallax; two infrared detectors can be used to triangulate points in the spatial coordinate plane of the operative field. This data can then be virtually superimposed or “registered” to the three-dimensional coordinate field of the reconstructed imaging model used for surgical planning. Additionally, a surgical probe can also be registered into this composite coordinate field to allow for precise iNav based on co-register of the points in the surgical field displayed on an intraoperative monitor. While these technologies necessarily distract from the surgical field, they represent the latest validated progression in personalized anatomical modeling and integrated surgical guidance (Table 1).

Surgical subspecialties with the smallest margins of error for surgical technique, such as neurosurgery, otolaryngology, and craniomaxillofacial surgery, benefit the most from the additional precision afforded by these simulations and are therefore understandably early technological adopters. As the first multidisciplinary pediatric head and neck surgical oncology clinic in the USA, we have found that the use of these technologies improves our operative confidence in the high stakes setting of pediatric head and neck cancer (pHNC). In fact, a combination of CASP, custom-manufactured anatomical surgical guides, and iNav allow our team the optimum opportunity to provide complete resections with clear margins and optimal approaches for aesthetic reconstruction. These technical optimizations are essential in the setting of increasing pHNC incidence, as investigated by our research group [11].

In this paper, we will review our center’s experience emphasizing the utility of iNav in our experience as well as report a scoping review of iNav’s application to pediatric head and neck oncology to date. By examining the existing literature for an evidence-based inclusion of iNav to pHNC surgical oncology in practice, we found sparse and fragmented reporting. Many studies were limited to lower levels of evidence, including case reports and small retrospective case series drawn from varied tumor types, anatomical sites, and surgical subspecialties. Their outcomes, marginal data, or case-specific data linkage were also inconsistently reported. Furthermore, no review was available for a collation of evidence across the literature; this represents an evidentiary gap in the pediatric surgical literature as to the application of iNav to the pediatric population. The intent of our work is to address this gap by collating the available evidence on the effect of iNav with regard to marginal control, approach planning, and morbidity reduction in pHNC. Specifically, we are interested in understanding: (1) the effect of iNav on the surgical endpoints (i.e., margins) and (2) the professional perception of both the utility and limitations of iNav in this setting. In keeping with our experience, we hypothesized that iNav would assist in achieving a margin-free resection and with concurrent affirmation by professional colleagues.

## 2. Materials & Methods

We performed a scoping review of the literature (PubMed, EMBASE, and Web of Sciences) for published reports of pHNC surgically resected with the assistance of iNav. We excluded central nervous system (CNS) and neurotological malignancies as well as benign craniofacial tumors. The review encompasses all studies with cases under the following inclusion criteria: pediatric patients (≤21), malignant or malignant-potential tumor in the head and neck region, surgically managed with the use of iNav for resection assistance. Our data collection categories included patient age, sex, tumor type, tumor location, length of follow-up period, resectional margin status, complications, and recurrence. We also collected and collated available statements by authors on the utility and limitations of iNav. Our search strategy is available for review in the attached Appendix A, and Figure 1 demonstrates our PRISMA diagram. All prospective studies obtained from the initial search were first abstract screened by two independent reviewers (J.W.; A.B.). A full text review was then conducted identically. Data extraction was performed by either reviewer, then data quality was confirmed by the second. The levels of evidence for clinical significance of the compiled cases were also collected. The amount of heterogeneity of iNav applications to pHNC precluded the possibility of a control group for feasible comparison of surgical outcomes in this population; therefore, we report no statistical analyses.

## 3. Results

Our scoping review identified 27 cases from 16 studies [12,13,14,15,16,17,18,19,20,21,22,23,24,25,26,27]. The average age of the patients was 10.7 years old (SD 5.1 yrs) with a M:F ratio of 1.9:1. Histologic diagnoses included Langerhans cell histiocytosis (8), chordoma (3), rhabdomyosarcoma (3), Ewing’s sarcoma (2), melanotic neuroectodermal tumor of infancy (2), osteoblastoma (2), osteosarcoma (1), low-grade mesenchymal tumor (1), germinoma (1), adenoid cystic carcinoma (1), metastasis of Ewing’s sarcoma (1), metastasis of melanoma (1), and metastasis of osteosarcoma (1). With regard to resection extent, 24/27 cases reported data with 18 achieving at least gross total resection (GTR) (i.e., R0) and 6 with subtotal/tumor debulking achieved (i.e., R1 and R2). Three cases (11, 21, and 26) recorded complications: transient diplopia (2), cranial nerve V3 anesthesia, and mild temporal trismus (Table 2). For the 22 cases reporting follow-up data, the mean follow-up duration was 2.27 years (SD 2.26). Twenty-three cases reported follow-up outcomes, which included no recurrence (18), local recurrence (2), distant metastasis (1), ongoing systemic oncologic therapy for concurrent metastasis (1), and oncologic mortality (1). Due to the heterogeneity of resection approaches and histopathology, statistically significant comparison of the utility of iNav was not feasible; however, Table 2 summarizes all of the case-specific details.

Many of the authors held favorable opinions regarding the utility of iNav technology, while concurringly concurrently agreeing upon its limitations in a pediatric setting. Table 3 and Table 4 summarize these narrative recommendations and qualifications of iNav, with which we also professionally concur.

## 4. Case Series

Five pHNC patients from our own surgical cohort underwent iNav-assisted surgery. Their case specific details are summarized in Table 5. In our practice, iNav is not universally applied. These cases were selected for iNav given the indications of high tumor invasiveness or prior history of operation or radiation history. These cases represent all utilization of iNav in our practice to date.

### 4.1. Case #1

A 10-year-old male with no relevant past medical history presented to our clinic with a visibly distorting mass of the nasal bridge/glabella. What was first assumed to be an insect bite was eventually pathologically diagnosed as a NUT-midline carcinoma. An initial complex radical resection requiring later multistage reconstructions was planned. Virtual surgical planningwith CAD/CAM was performed pre-operatively with the production of multiple surgical guiding models. In the perioperative period, the patient’s facial features were registered using the standard operation of a Stryker Navigation System II (Stryker Inc., Kalamazoo, MI, USA), and an outline of the widest marginal dimensions was drawn on the facial surface to outline the necessary incisions. After en-bloc resection, the intraoperative navigation probe was used to compare the depth of resection (i.e., cribriform plate) to the surgical plan and to identify appropriate locations for intraoperative frozen sections for pathological diagnosis. Intraoperative imaging and model-based guidance were also used to assist in transfer of the operative plan in this high-stakes operation (Figure 2).

### 4.2. Case #2

A 12-year-old male with a history of mucoepidermoid carcinoma (MEC) treated with parotidectomy and radiotherapy at an outside center presented to our clinic for cancer surveillance care. An annual head and neck MRI revealed a new mass in the surgical bed, which led to a fine-needle aspiration (FNA) biopsy confirming recurrence of MEC. Surgical excision with selective neck dissection was planned with use of iNav. Perioperatively, the Stryker CranialMap (Stryker Inc., Kalamazoo, MI, USA) was used to register patient’s facial anatomy with the preoperative imaging. Next, the surgical navigation probe was used to mark out sufficient margins on the skin surface for resection due to tumor proximity to the skin. The primary tumor resection and a selective neck dissection was performed with difficulty due to cicatricial tissue as a result of prior radiotherapy and invasion of the facial nerve, but iNav afforded the surgical team the capability to achieve an R0 resection (Figure 3).

### 4.3. Case #3

A 17-year-old male re-presented to our clinic after endoscopic biopsy of a nasopharyngeal mass, diagnosed as Burkitt’s lymphoma. Chemotherapy led to a rapid clinical response, but the nasopharyngeal mass was not resolved. Repeat endoscopic biopsy was indicated to confirm complete chemotherapeutic response. After a standard registration process perioperatively, the patient underwent endoscopic nasopharyngeal examination. The intraoperative procedure allowed for exact placement of the biopsy location based on preoperative imaging. Final pathology indicated complete response to chemotherapy (Figure 4).

### 4.4. Case #4

A 15-year-old female originally presented to our clinic for excision of a right parotid mass presumed to a branchial cleft cyst. Surgical resection was aborted due to intraoperative frozen biopsy findings suggestive of low-grade malignancy. Permanent pathology results revealed acinic cell carcinoma requiring definitive oncologic resection. Preoperative imaging, including PET scan, showed no evidence of metastatic disease, while MRI demonstrated the lesion confined to the parotid tail with well-defined margins. Following multidisciplinary discussion at tumor board, the patient underwent oncologic resection of the right parotid tumor combined with a selective neck dissection, utilizing iNav. During the procedure, standard registration was performed using the Stryker CranialMap (Stryker Inc., Kalamazoo, MI, USA) to confirm the precise location of the tumor and its margins as demonstrated on preoperative imaging. The surgical probe was utilized to mark 1 cm margins on the skin surface, encompassing portions of the sternocleidomastoid and digastric muscles. The dissection proved to be challenging due to the patient’s prior surgery and involvement of the main marginal mandibular branch of the facial nerve, which necessitated nerve reconstruction. The assistance of iNav ultimately facilitated a complete R0 resection, as confirmed by final pathology (Figure 5).

### 4.5. Case #5

A 17-year-old female was referred to our clinic for evaluation of a right parotid gland mass, which demonstrated low-grade acinic cell proliferation on fine-needle aspiration (FNA), suspicious for acinic cell carcinoma. After discussion at tumor board, the patient underwent a total right parotidectomy with selective neck dissection utilizing iNav. Following standard registration, the iNav surgical probe was used to localize the tumor and delineate its margins, which were marked on the skin. The anticipated course of the facial nerve branches was then mapped relative to the tumor. The lesion was then resected en bloc via total parotidectomy. Dissection necessitated reconstruction of the marginal, buccal, zygomatic, and frontal branches of the facial nerve due nerve invasion by the tumor. Final pathology confirmed acinic cell carcinoma with negative margins (Figure 6).

## 5. Discussion

Stereotactic surgery has been in common practice for neurosurgery and otolaryngology since its commercial introduction in the 1990s [28]. Craniomaxillofacial surgery also represents an ideal application of this technology, with ongoing descriptions in the literature. For example, iNav has proven utility in orbital reconstruction [29,30], benign tumor resection [31,32], and dental implantation [33]. Thus, conceptually, the surgical management of head and neck cancer represents a natural extension of this technology’s application, due to the necessity of surgical precision in achieving an R0 resection [34,35,36], with a recent meta-analysis suggesting its potential as a standard of care in adult surgical oncology [37]. In sinonasal carcinomas, tumors impinging on the deep facial compartments, and the anterior skull base, iNav is integrated with endoscopic resection, with increased safety and surgeon confidence due to an understanding of patient-specific anatomy [38,39]. In tumor-resecting maxillectomies, preoperative planning using iNav significantly improved negative surgical margins [40,41,42].

As evidenced by our scoping review, the published applications of iNav in the setting of pHNCs present lower-level evidence, such as single cases and retrospective cohort studies. However, no authors discouraged its use, and most commented favorably concerning its helpfulness as a surgical tool, despite generally remediable limitations. While these lower-level sources of evidence may seem to hinder the evidentiary support of iNav, this would be a premature conclusion. If a medical specialty has already incorporated a treatment adjunct into clinical practice, to randomize a patient to a treatment without iNav would be to perform non-indicated treatment, which represents an ethical prohibition [27,43]. Experiential expert opinion forms sufficient basis for employing iNav as an indicated treatment adjunct in this setting. However, continued research contribution to the literary paucity in this area would greatly benefit the professional understanding of this technology and contribute to the existing knowledge base. This was the intention of our scoping review in collating all this relevant evidence for professional consideration.

Our study has several important limitations. Study inclusion and data extraction for the scoping review was difficult due to generally incidental reporting on the use of iNav. There were several papers that included both mention of iNav and at least one pHNC; however, there was insufficient evidential justification for their inclusion. Frequently, these papers were extensive institutional series without data linkage between case specific details. Alternatively, sufficient linkage existed, yet without explicit description of iNav utilization. Additionally, some publications utilized iNav in single-stage resection and reconstruction procedures; these were excluded when it was unclear or reasonably evident that the iNav was largely applied for the reconstructive intention. Finally, our institutional case series examined a two-year study period from 2023–2025. The earliest published study included in our scoping review was published in 2004 [11]. This two-decade discrepancy limits direct comparisons, as iNav technology significantly improved during that time, including imaging resolution, computer reconstructions, intraoperative integration, etc. This represents a limitation with effective data comparison. Additionally, our case series is of a minimal sample size with limited utility in comparison of iNav techniques.

In our pediatric head and neck surgical oncology practice, iNav provides a number of benefits, in concurrence with the professional consensus. In combination with established preoperative planning methods, iNav provides tumor margin delineation to trace the initial surgical approach in resective surgery. Intraoperative assessment of the depth of the resection bed compared to lesional depth on imaging is also made possible using this technology. Finally, iNav has also been shown to enable selection of appropriate biopsy sites. These advantages, which are common to our adult counterparts, are of potentially greater clinical importance to the pediatric population, due to smaller, more anatomically concentrated operative fields, which may be mapped out using iNav. Additionally, the implications of achieving a negative-margin (R0) resection in pediatric oncology have a weightier significance, given the four-fold difference in average potential years of life lost due to cancer deaths in pediatric cancer vs. adult oncology [44,45].

Yet, our scoping review still found a high rate of R1/R2 resections. This analysis reveals that technology can never be a replacement for operative clinical judgement and technique. Even a perfected integration of perioperative imaging data into the surgical environment would be no substitute or guarantee of microscopic pathological extent. It is likely that dedifferentiated, highly invasive tumors, as well as those in anatomically concentrated regions, are at higher risk of a subtotal resection. Indeed, both the literature and our case series demonstrate that computer-assisted navigation does not eliminate the risk of positive margins nor morbidity from oncologic nerve sacrifice or complex dissection. The value of iNav in this setting continues to be supportive rather than determinative.

The future of computer-assisted navigation technology for the purpose of surgical oncology is bright with multiple avenues for further integration and technological development. Stereotactic navigation would be benefitted by further integration between the surgical modeling and the intraoperative field. This is being actively pursued in at least three dimensions. A current limitation is the displacement of the display of digital modeling on a computer display out of the operative field of view; this same problem is resolved in military applications using a heads-up display (HUD). Integrating solutions currently being tested include image projections into the operative field [46] and the use of a HUD projected through commercially available augmented/mixed-reality platforms [47,48,49,50]. The implementational difficulties of these technologies include the difficulty of a secondary registration (parallax error) of a projected surgical model back into the surgeon’s field of view [47]. The second dimension of stereotactic surgical improvement is the additional registration of surgical instruments into the digital modeling environment, further enhancing the surgeon’s operative understanding of the potential effect of a surgical trajectory in relation to concealed critical anatomy [51,52,53]. For example, rather than only a registered surgical probe, registration of an osteotome would allow for the real-time digital modeling of the osteotomy vector and tumoral margin, confirming correct surgical approach and allowing for micro-adjustments [54,55]. Finally, optically-based navigation systems introduced the technology; however, they have several drawbacks, including the necessity of a shallow surgical field, a direct line of sight to the parallax optical detector, and the limitation of co-registered instruments due to ungainly optical reflection attachments [56]. Therefore, magnetic coordinate navigation represents a more ideal tracking technology for the purposes of intraoperative navigation in surgical oncology [57,58,59,60,61].

Stereotactic surgical robotic assistance is also a technological development with potential for operative navigation enhancement [62,63]. After a primary registration to the patient’s operative anatomy, the robotic arm may be used to maintain precise vector and planes for instrumentation, potentially even with programmed mechanical hard-stops to prevent deviation or overcuts [64,65,66,67]. Another technologic development with navigational potential would be improved imaging with further resolution for the purpose of neurovascular avoidance. Currently, the imaging data obtained as a result of standard MRI and CT scans does not have sufficient resolution or tissue contrast to reliably distinguish the minute diameter of the extra-cranial segments of the cranial nerves. However, with the development and adoption of higher magnetic force MRI scanners [68], innovative MRI sequences for tissue contrast [69,70], and automated segmentation protocols to define these minute regions of interest (ROI) [71,72,73], these critical structures could be routinely included in presurgical and intraoperative modeling for improved preservation of these critical nerves as they cross the operative field in complex craniofacial resection [74,75,76].

## 6. Conclusions

Intraoperative navigation (iNav) represents a surgical technology with low to reasonable utilization as well as being an active surgical research topic. While iNav demonstrates proven clinical utility in adult applications, its application in the setting of pediatric head and neck cancer (pHNC) presents significant potential for surgical optimization for these difficult cases. Additional research will likely continue to demonstrate its clinical utility; however, iNav is presently indicated by common practice and professional adoption into clinical experience, though not formal consensus guideline recommendations. In the field of pediatric surgical oncology, where the balance of thorough extirpation and minimized surgical morbidity is essential, iNav has significant utility.

## Figures and Tables

**Figure 1 cancers-18-00154-f001:**
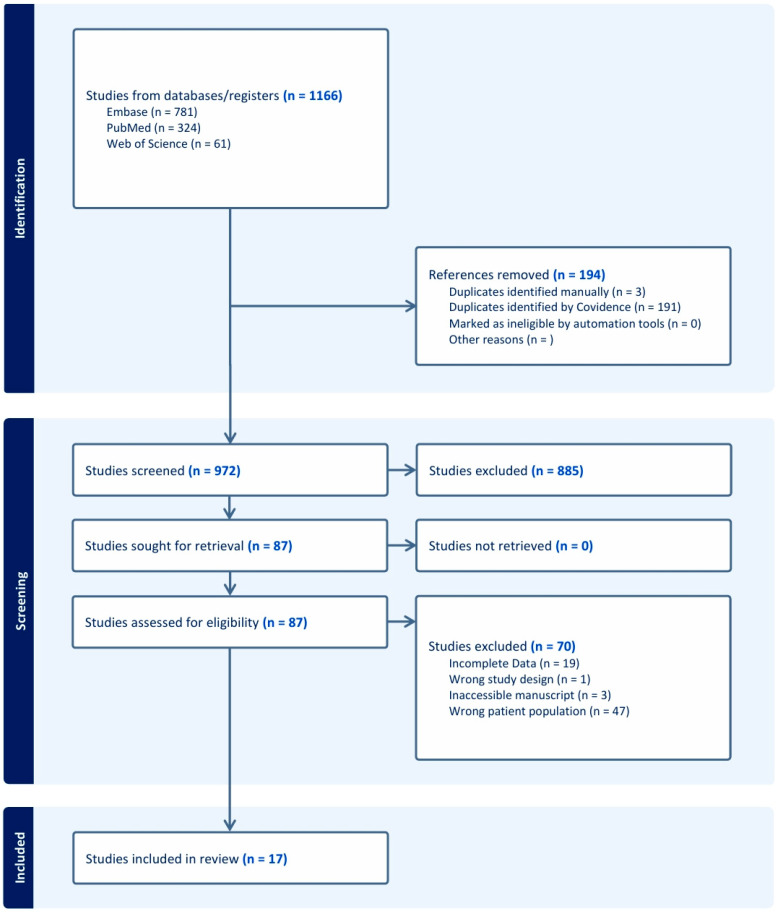
PRISMA diagram for our review.

**Figure 2 cancers-18-00154-f002:**
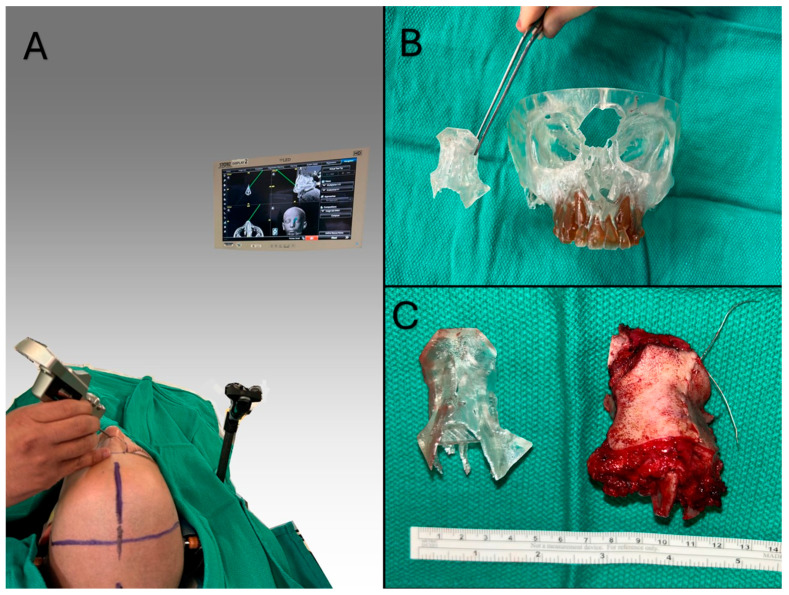
(**A**): A simplified intraoperative view demonstrating the perioperative use of navigation probe to mark out the edge of the nasal resection margin. (**B**): The intraoperative surgical guide showing the detachable element representing the optimal resection specimen. (**C**): The resection specimen compared directly with this detachable resection model.

**Figure 3 cancers-18-00154-f003:**
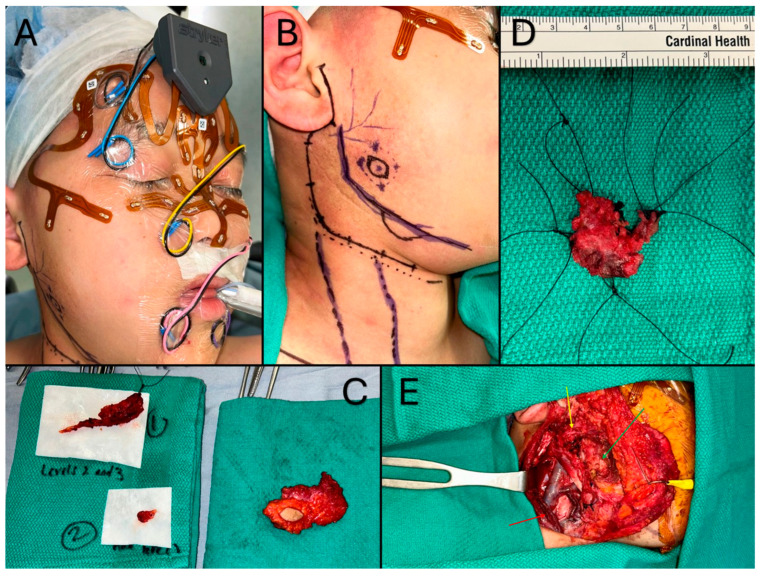
(**A**): Perioperative view demonstrating CranialMap placement for registration of facial surface as well as electrode placements in the end effectors of the facial nerve for neurophysiological monitoring. (**B**): Perioperative view of incisional outlines, critical anatomical features, and the critical margins of the mass. (**C**): Surgical specimens of the initial mandibular mass with intact skin margin (right) and lymphovascular package specimens (left). (**D**): View of the radical resection specimen with sternocleidomastoid margin after pathological margin positivity. (**E**): Surgical bed after radical excision showing edge of resected sternocleidomastoid (red arrow), preserved frontal and zygomatic branches of the facial nerve (yellow arrow), and mandibular surface after masseteric excision (green arrow).

**Figure 4 cancers-18-00154-f004:**
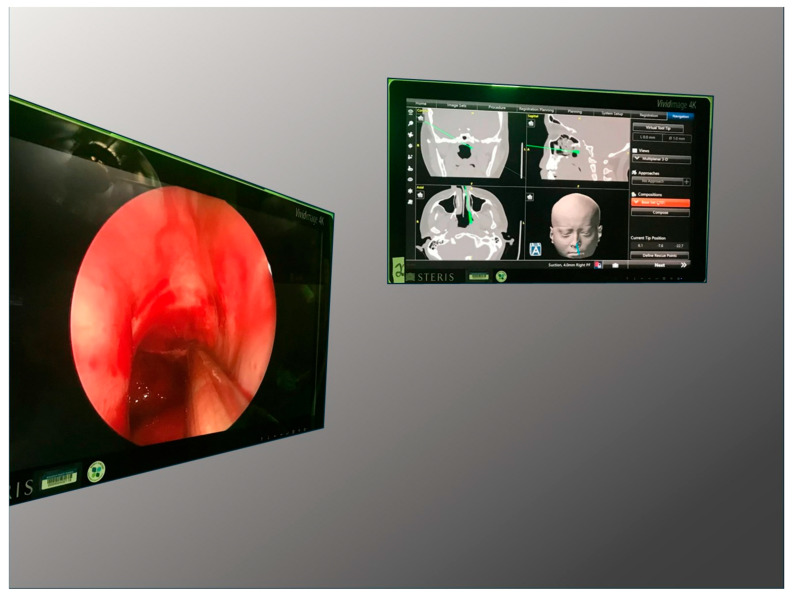
Simplified intraoperative view showing the endoscopic use of navigation for ideal localization of the biopsy location.

**Figure 5 cancers-18-00154-f005:**
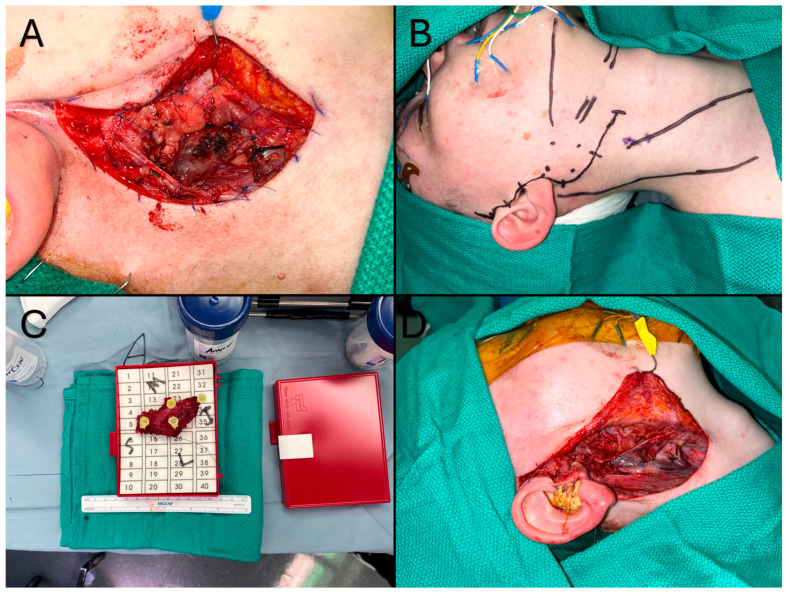
(**A**): This figure demonstrates the surgical bed during the initial aborted operation where malignancy was discovered via frozen pathology. Sutures mark the presumed margin for return definitive oncologic resection. (**B**): This figure demonstrates the preoperative layout and skin markup for complete oncologic resection. The four dots indicate the medial, lateral, superior, and inferior margins of the tumor, as identified by intraoperative navigation (iNav). (**C**): This figure demonstrates the resected tumor with margin. Resection of this tumor required sacrifice of a branch of the facial nerve. The Superior (S), Lateral (L), Inferior (I), and Medial (M) margins of the Specimen were marked for the pathologist. (**D**): This figure demonstrates the post-resection surgical field, specifically the facial nerve branch repair.

**Figure 6 cancers-18-00154-f006:**
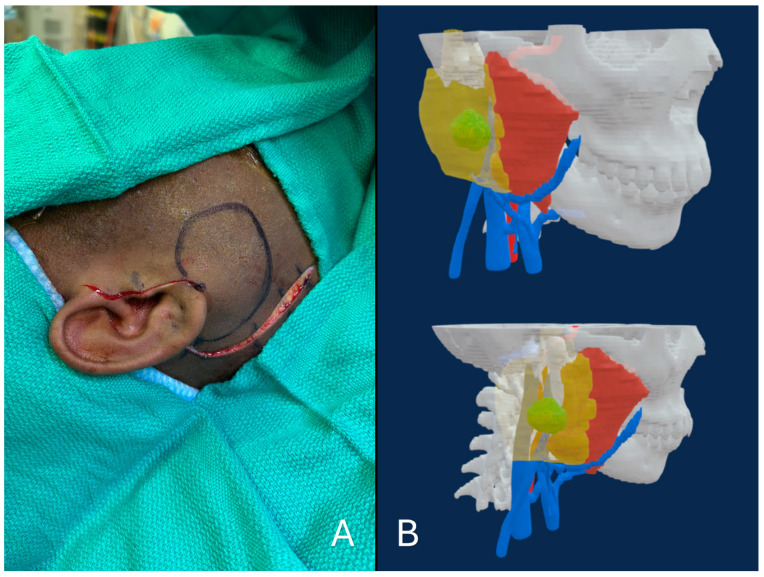
(**A**): This figure demonstrates the identified tumor margins by intraoperative navigation (iNav). (**B**): This figure demonstrates two views of the 3D rendering of the tumor model. This model was 3D printed for use as a stereolithic surgical aid. The lime-coloured mass within the transparent yellow mass represents the tumor within the parotid gland while the blue linear masses represent venous vasculature, and the red mass represents the masseter muscle.

**Table 1 cancers-18-00154-t001:** Navigational technologies compared according to relevant characteristics.

Navigational Technology	Type	Guidance	Limitations
Intraoperative imaging (CT, MRI, fluoroscopy, etc.)	Comparative	Intraoperative computerized visualization of surgical intervention	No comparison within the virtual planning environment Expensive infrastructure and complex workflows
3D-printed surgical models	Comparative	Intraoperative modeling for ideal surgical progression. Personalized anatomical model	Difficult mapping of the landmarks intraoperatively Plastic model is a rough approximation of human tissue
Intraoperative navigation	Integrative	Intraoperative computer-assisted navigation in the surgical field using preoperative imaging High resolution (margin of error less than 1 mm)	Requires a display monitor, distracting from the surgical field Increased anesthesia time for perioperative registration process
Augmented reality	Integrative	Idealized intraoperative integration of virtual surgical plan directly on to the surgical field of view	Preclinical development phase with unresolved technical challenges, such as integrating a heads-up display with surgical loupes

**Table 2 cancers-18-00154-t002:** Case-specific details obtained as a result of our scoping review of the literature.

Case Number	Age	Sex	Diagnosis	Anatomic Location	Resectional Status	Follow Up Length	Outcome	Complications
1	4	Male	LCH	Parietal calvarium	GTR	10 years	no recurrence	None
2	2	Female	LCH	Frontal calvarium	GTR	1 years	local recurrence	None
3	3	Female	LCH	Frontal calvarium	GTR	4 month	no recurrence	None
4	14	Male	Melanoma metastasis	Occipital calvarium	STR	1 years	no recurrence	None
5	8	Male	Low grade mesencyhmal tumor	Frontal calvarium	GTR	5 months	no recurrence	None
6	11	Male	Osteoblastoma	Occipital calvarium	GTR	18 months	no recurrence	None
7	8	Male	LCH	Frontal calvarium	GTR	4 years	no recurrence	None
8	6	Male	LCH	Temporal calvarium	GTR	6 years	local recurrence	None
9	8	Female	Rhabdomyosarcoma	Skull base and parapharyngeal space	GTR	ND	ND	None
10	7	Female	Rhabdomyosarcoma	Pterygomandibular space	GTR	ND	ND	None
11	8	Male	Osteoblastoma	Fronto-temporal-orbital calvarium	ND	3 years	no recurrence	Transient diplopia
12	16	Male	Ewing’s Sarcoma	C2 vertebra and prevertebral space	STR	1 year	no recurrence	None
13	14	Male	Chordoma	C2 vertebra and bilateral masses	STR	3 months	no recurrence	None
14	20	Male	Osteosarcoma metastasis	Epidural and paraspinal spaces near cervical vertebrae	GTR	14 months	concurrent lung metastasis	None
15	9	ND	LCH	C2 cervical vertebra	STR	ND	ND	None
16	18	Female	Adenoid cystic carcinoma	Skull base and infratemporal space	GTR	30 months	distant metastasis	None
17	7	Male	Ewing’s sarcoma metastasis	Parietal calvarium	ND	10 months	Death due to treatment resistant Ewings	None
18	12	Male	Ewing sarcoma	Temporal calvarium	GTR	16 months	no recurrence	None
19	16	Female	Chordoma	Skull base, craniocervical junction, prevertebral space, and bilateral invasion	STR	33 months	no recurrence	None
20	14	Male	Chordoma	Skull base, craniocervical junction, prevertebral space, and bilateral invasion	GTR	31 months	no recurrence	None
21	2	Male	Germinoma	Inferomedial orbital space	STR	ND	no recurrence	Transient diplopia
22	8	Male	Melanotic neuroectodermal tumor of infancy	Parietotemporal calvarium	GTR	2 years	no recurrence	None
23	2 months	Female	Melanotic neuroectodermal tumor of infancy	Maxilla	GTR	1 years	no recurrence	None
24	11	Male	LCH	Frontal calvarium	GTR	1.5 year	no recurrence	None
25	7	Male	LCH	Frontal calvarium	GTR	2 years	no recurrence	None
26	16	Female	Osteosarcoma	Pterygoidal, nasopharyngeal, infratemporal, and parapharyngeal spaces	GTR	1 years	no recurrence	Anesthesia of the left V3 distribution and mild temporary trismus
27	10	Female	Rhabdomyosarcoma	Infratemporal space	ND	ND	ND	None

**Table 3 cancers-18-00154-t003:** Qualitative summary statement of author’s consensus concerning the utility of iNav with the level of evidence represented.

Utility Statement	Supporting Literature	Level of Evidence
iNav improves surgical confidence and orietation with large complex tumors, altered anatomy such as cicatricial tissue, or dissection of critical structures.	[13]	4
	[14]	5
	[16]	5
	[17]	4
	[23]	5
	[27]	4
iNav assists the surgeon in minimizing unnecessary surgical morbidity through avoidance of surgical margin over-resection as well as potential excess radition from intraoperative imaging for navigation.	[12]	4
	[17]	4
	[19]	4
	[24]	5
	[27]	4
iNav is beneficial in achieving surgically clear margins, especially for neoplasm where margins are curatively critical, as evidenced by low recurrence rates.	[12]	4
	[24]	5
	[27]	4
iNav is beneficial in avoiding surgical complications.	[12]	4
	[27]	4

**Table 4 cancers-18-00154-t004:** Qualitative summary statement of author’s consensus concerning the limitations of iNav with the level of evidence represented.

Limitation Statement	Supporting Literature	Level of Evidence
iNav is limited by a procedural assumption of concrete tissue and permanent anatomy. This is confronted by inherent surgical modification/distortion of the anatomy, the natural reality of plastic/moldable tissues, developing/evolving anatomy in pediatric patients, and potentially disparate positioning for imaging vs. surgery. Authors propose various solutions to assist in ameliorating this issue.	[13]	4
	[16]	5
	[17]	4
	[22]	4
	[24]	5
Meticulous, initual registration of the iNav system is critical for effective use, due to the impossibility of re-registration intraoperatively. This requirement increases operative length as well as skill maintanence requirements by providers.	[13]	4
	[14]	5
	[24]	5
	[25]	5
iNav is technologically complex, involving an additional system of navigation hardware. These may be anchored in the surgical field or to the patient directly, requiring extreme operative caution and negatively impacting surgical ergonomics.	[14]	5
iNav, as an adjunctive surgical technology, is an expensive addition to an already complex surgery.	[24]	5

**Table 5 cancers-18-00154-t005:** Institutional case details concerning iNav utilization in pHNC patients.

Series #	Age	Sex	Diagnosis	Anatomic Location	Resectional Status	Follow-Up Length	Outcome To date	Complications
1	10	M	NUT Carcinoma	Glabella	GTR	13 months	No recurrence	L vision loss due to partial tarsorrhaphy
2	12	M	MEC	Parotid	GTR	22 months	No recurrence	CN7 Palsy due to sacrifice and reconstruction
3	17	M	Burkitt’s Lymphoma	Nasopharyngeal	N/A	28 months	No recurrence	None
4	15	F	Acinic Cell Carcinoma	Parotid	GTR	9 months	No recurrence	CN7 Palsy due to sacrifice and reconstruction
5	17	F	Acinic Cell Carcinoma	Parotid	GTR	7 months	No recurrence	CN7 Palsy due to sacrifice and reconstruction

## Data Availability

The original contributions presented in this study are included in the article/Appendix A. Further inquiries can be directed to the corresponding author.

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
