# Peer review of "Computer-Assisted Intraoperative Navigation in Pediatric Head and Neck Surgical Oncology: A Single-Center Case Series and Scoping Review of the Literature"

_cancers, 2026, doi:10.3390/cancers18010154_

Round 1
Reviewer 1 Report
Comments and Suggestions for Authors
This is a very nice case presentation of the current possibilities as well as gives an overview of the existing literature.
In the abstract we start of with a bold claim saying pediatric cancer is increasing, I am not too sure that's actually true according to the stats of CCLG and COG so either add a reference or temper this statement, furthermore this is rot repeated in the introduction what so ever, needs to be taken out.
The introduction is quite lengthy and a gap in pediatric oncology should be identified a bit more thoroughly.
For the case series presentation:
- it would be nice to clarify why these 5 cases were picked
- in the table surgical approach and reconstruction is left out, this would need to be added
- were any cases handled without iNav - should be described, or is this your common practice?
- if so what are your indications for iNav vs non I Nav
- morbidity is not conceptualized.
- The authors claim a 100% R0 resection yield; if so I think clarification is needed as to case selection, nerve sacrifice and late effects.
The discussion is diffuse and does not necessarily directly link into the presented data. Furthermore, the discussion is quite speculative and takes away from the finding, some nuance would help.
The conclusions should be tempered. Statements such as “iNav is presently indicated by professional consensus” and “ideal opportunity for this surgical adjunct” overstate the findings. Giving its small sample size, potential selection bias, heterogeneity, absence of comparison, low level of evidence in literature.
Author Response
Comment 1: "This is a very nice case presentation of the current possibilities as well as gives an overview of the existing literature." Response 1: We appreciate the kind compliment and look forward to the opportunity to contribute to the published literature.
Comment 2: "In the abstract we start of with a bold claim saying pediatric cancer is increasing, I am not too sure that's actually true according to the stats of CCLG and COG so either add a reference or temper this statement, furthermore this is rot repeated in the introduction what so ever, needs to be taken out." Response 2: Thank your for this comment. This statement was made in relation to the published works of our own group as well as other authors. We have cited the paper in question and have made the following amendments to the manuscript to provide better evidence for this claim: "These technical optimizations are essential in the setting of increasing pHNC incidence, as investigated by our research group [11] (pg 3, fifth paragraph of the Introduction).
Comment 3: "The introduction is quite lengthy and a gap in pediatric oncology should be identified a bit more thoroughly." Response 3: Thank you for this comment. We intended to provide a comprehensive introduction to the breadth of surgical adjuncts to contextualize our work on iNav for less familiar readership. We have included the following additions to the revised manuscript to further identify the literary gap that we hope to begin to fill with this work: " In examining the reported literature for an evidence-based inclusion of iNav to our pHNC surgical oncology practice, we found sparse and fragmented reporting. Many were limited to lower levels of evidence, including case reports and small retrospective case series drawn from varied tumor types, anatomical sites, and surgical subspecialties. Studies inconsistently reported outcomes, marginal data, or case-specific data linkage. Further, no review was available for a collation of literature evidence; this represents an evidentiary gap in the pediatric surgical literature as to the application of iNav to the pediatric population. This is the intent of our work: to collate the available evidence as to the effect of iNav on marginal control, approach planning, or morbidity reduction in pHNC. (pg 3, sixth paragraph of the Introduction)"
Comment 4: "For the case series presentation:
- it would be nice to clarify why these 5 cases were picked
- in the table surgical approach and reconstruction is left out, this would need to be added
- were any cases handled without iNav - should be described, or is this your common practice?
- if so what are your indications for iNav vs non I Nav
- morbidity is not conceptualized.
- The authors claim a 100% R0 resection yield; if so I think clarification is needed as to case selection, nerve sacrifice and late effects. "
Response 4: We appreciate the improvement that these questions bring to the quality of our manuscript. We have included a summary table of our case series and additional information concerning case selection and our use of iNav in our institutional practice. " Five pHNC patients from our own surgical cohort underwent iNav-assisted surgery. Their case specific details are summarized in Table 5. In our practice, iNav is not universally applied. These cases were selected for iNav given the indications of high tumor invasiveness or prior history of operation or radiation history. These cases represent all utilization of iNav in our practice to date.
|
Table 5 |
|
|
|
|
|
|
|
|
|
Series # |
Age |
Sex |
Diagnosis |
Anatomic Location |
Resectional Status |
Follow-up Length |
Outcome To date |
Complications |
|
1 |
10 |
M |
NUT Carcinoma |
Glabella |
GTR |
13 months |
No recurrence |
L vision loss due to partial tarsorrhaphy |
|
2 |
12 |
M |
MEC |
Parotid |
GTR |
22 months |
No recurrence |
CN7 Palsy due to sacrifice and reconstruction |
|
3 |
17 |
M |
Burkitt’s Lymphoma |
Nasopharyngeal |
N/A |
28 months |
No recurrence |
N/A |
|
4 |
15 |
F |
Acinic Cell Carcinoma |
Parotid |
GTR |
9 months |
No recurrence |
CN7 Palsy due to sacrifice and reconstruction |
|
5 |
17 |
F |
Acinic Cell Carcinoma |
Parotid |
GTR |
7 months |
No recurrence |
CN7 Palsy due to sacrifice and reconstruction |
(pg 5, First paragraph of the Case Series Section)
Comment 5: "The discussion is diffuse and does not necessarily directly link into the presented data. Furthermore, the discussion is quite speculative and takes away from the finding, some nuance would help." Response 5: We appreciate the reviewer's concerns and have included the following language to improve the nuance of the discussion: " Indeed, the literature and our case series show that navigation does not eliminate the risk of positive margins nor morbidity from oncologic nerve sacrifice or complex dissection. The value of iNav in this setting continues to be supportive rather than determinative. (Pg. 12, paragraph 5 of the Discussion).
Comment 6: "The conclusions should be tempered. Statements such as “iNav is presently indicated by professional consensus” and “ideal opportunity for this surgical adjunct” overstate the findings. Giving its small sample size, potential selection bias, heterogeneity, absence of comparison, low level of evidence in literature." Response 6: We appreciate the value that this comment brings to our manuscript. By using the language "professional consensus" we did not intend to imply an indication level of "professional consensus guidelines". We appreciate the potential for semantic overlap of these two phrasings and have changed the language to "indicated by common practice and professional adoption into clinical experience, though not formal consensus guideline recommendations. In this field of pediatric surgical oncology where the balance of thorough extirpation and minimized surgical morbidity is essential, iNav has significant utility. (pg 15, Conclusion)
Reviewer 2 Report
Comments and Suggestions for Authors
This manuscript presents a timely and important investigation into the application of intraoperative navigation (iNav) in pediatric head and neck cancer surgery. The combination of a scoping review of the existing literature and a single-center case series review provides some insight into a field where published applications are low-level evidence. The methodology is sound, I have no issues regarding internal or external validity.
The justification for relying on lower-level evidence due to the ethical prohibitions associated with withholding an established surgical adjunct in high-stakes oncology cases is relevant in assembling the manuscript.
The authors identify a gap in the published literature concerning iNav application in the pediatric population. The rationale for performing a scoping review is supported by the recognized heterogeneity of iNav applications to pHNC, which precludes statistical analyses and control group comparisons for surgical outcomes. It is important that the authors clearly state that while iNav is presently indicated by professional consensus, the current published evidence base primarily consists of single cases and retrospective cohort studies.
The scoping review identified 27 cases from 16 studies. Of the 24 cases reporting resection extent, 18 achieved R0 margins, while 6 achieved R1 and R2 margins. The authors note that the review still found a high rate of R1/R2 resections, confirming that technology cannot replace operative clinical judgment and technique. This is a crucial finding that should be highlighted further, as it is an important issue regarding iNav's utility.
A significant discrepancy exists in the reported timelines. The scoping review was performed according to PRISMA guidelines from 1970 to present. The single-center retrospective case series covered experience from September 2022 to September 2025, comments on possible time-related differences in using iNav are necessary.
The five-case series is cited as achieving a 100% R0 resection rate with the use of iNav. This outcome strongly supports the institutional conclusion that iNav facilitates marginal assessment, guides the surgical approach, and confirms the depth of resection. However, it should be clarified that case #3 was an endoscopic nasopharyngeal examination where iNav was used for exact placement of a biopsy location to confirm chemotherapeutic response, rather than a definitive tumor resection for margin status. This distinction is important for interpreting the R0 rate applied to the remaining four definitive resection cases (Cases 1, 2, 4, 5). The authors correctly affirm that iNav has been shown to enable the selection of appropriate biopsy sites.
The Discussion section thoroughly outlines future technological advancements, and discusses relevant literature.
The authors note that study inclusion and data extraction for the scoping review were difficult due to incidental reporting and insufficient evidential justification, leading to the exclusion of extensive institutional series without specific case data linkage or explicit iNav utilization details. This is an important limitation to underscore in the Abstract or conclusion regarding the breadth of the current literature review.
Author Response
Comment 1: "This manuscript presents a timely and important investigation into the application of intraoperative navigation (iNav) in pediatric head and neck cancer surgery. The combination of a scoping review of the existing literature and a single-center case series review provides some insight into a field where published applications are low-level evidence. The methodology is sound, I have no issues regarding internal or external validity." Response 1: We thank Reviewer 1 for their compliments.
Comment 2: "The justification for relying on lower-level evidence due to the ethical prohibitions associated with withholding an established surgical adjunct in high-stakes oncology cases is relevant in assembling the manuscript." Response 2: We concur with Reviewer 1's assessment of the ethical implications of technological adoption in this setting.
Comment 3: "The authors identify a gap in the published literature concerning iNav application in the pediatric population. The rationale for performing a scoping review is supported by the recognized heterogeneity of iNav applications to pHNC, which precludes statistical analyses and control group comparisons for surgical outcomes. It is important that the authors clearly state that while iNav is presently indicated by professional consensus, the current published evidence base primarily consists of single cases and retrospective cohort studies. Response 3: The author concur with this concern. However, given other feedback we have modified our original wording of "indicated by professional consensus" due to a potential semantic overlap with "professional consensus guidelines". We have changed this language to "indicated by common practice and professional adoption into clinical experience" (page 15, Conclusion paragraph).
Comment 4: " The scoping review identified 27 cases from 16 studies. Of the 24 cases reporting resection extent, 18 achieved R0 margins, while 6 achieved R1 and R2 margins. The authors note that the review still found a high rate of R1/R2 resections, confirming that technology cannot replace operative clinical judgment and technique. This is a crucial finding that should be highlighted further, as it is an important issue regarding iNav's utility." Response 4: We appreciate the connection suggested by this comment. We have update the manuscript to reflect this by adding additional commentary following this explanation of a high rate of R1/R2 resections. "Even a perfected integration of peri-surgical imaging data into the surgical environment would be no substitute or guarantee of microscopic pathological extent. It is likely that dedifferentiated, highly invasive tumors as well as those in anatomically concentrated regions are at higher risk of a subtotal resection (pg. 12, Paragraph 3)".
Comment 5: "A significant discrepancy exists in the reported timelines. The scoping review was performed according to PRISMA guidelines from 1970 to present. The single-center retrospective case series covered experience from September 2022 to September 2025, comments on possible time-related differences in using iNav are necessary" Response 5: We appreciate this comment and thank Reviewer 1 for bringing it to our attention. We have added the following sentences in commentary: " Finally, our institutional case series examined a two year study period from 2023-25. The earliest published study included in our scoping review was published in 2004 [11]. This two decade discrepancy limits direct comparisons, as iNav technology significantly improved during that time, including imaging resolution, computer reconstructions, intraoperative integration, etc. This represents a limitation with effective data comparison. (pg. 12, Paragraph 1)"
Comment 6: " The five-case series is cited as achieving a 100% R0 resection rate with the use of iNav. This outcome strongly supports the institutional conclusion that iNav facilitates marginal assessment, guides the surgical approach, and confirms the depth of resection. However, it should be clarified that case #3 was an endoscopic nasopharyngeal examination where iNav was used for exact placement of a biopsy location to confirm chemotherapeutic response, rather than a definitive tumor resection for margin status. This distinction is important for interpreting the R0 rate applied to the remaining four definitive resection cases (Cases 1, 2, 4, 5). The authors correctly affirm that iNav has been shown to enable the selection of appropriate biopsy sites." Response 6: Again, we appreciate this comment and thank Reviewer 1 for bringing it to our attention. We have clarified the abstract with this revision: "The case series review identified five cases of pHNC that met inclusion criteria. This small case series revealed a 100% RO resection rate with the use of iNav in four pHNC resections. A fifth case utilized iNav in biopsy site selection." (pg. 2, Abstract: Results subsection)
Comment 7: " The Discussion section thoroughly outlines future technological advancements, and discusses relevant literature.
The authors note that study inclusion and data extraction for the scoping review were difficult due to incidental reporting and insufficient evidential justification, leading to the exclusion of extensive institutional series without specific case data linkage or explicit iNav utilization details. This is an important limitation to underscore in the Abstract or conclusion regarding the breadth of the current literature review. Response 7: We appreciate the value of this comment and have modified the Conclusion subsection of our abstract as follows: " This study was limited by incidental and incomplete reporting of iNav’s clinical application to pHNC; several extensive institutional reports had to be excluded due to insufficiently detailed data linkage (pg. 2, Abstract: Conclusion subsection)"